# From Loss of Control to Social Exclusion: ERP Effects of Preexposure to a Social Threat in the Cyberball Paradigm

**DOI:** 10.3390/brainsci12091225

**Published:** 2022-09-10

**Authors:** Xu Fang, Yu-Fang Yang, Rudolf Kerschreiter, Michael Niedeggen

**Affiliations:** 1Division of Experimental Psychology and Neuropsychology, Department of Education and Psychology, Freie Universität Berlin, 14195 Berlin, Germany; 2Division of Social, Organizational, and Economic Psychology, Department of Education and Psychology, Freie Universität Berlin, 14195 Berlin, Germany

**Keywords:** Cyberball, social exclusion, loss of control, preexposure, expectancy violation

## Abstract

Previous studies indicated that the onsets of different social threats, such as threats to ”belonging” and “control”, are inconsistent with the subjective beliefs of social participation and require readjustment of expectations. Because a common cognitive system is assumed to be involved, the adjustment triggered by the experience of a single social threat should affect the processing of subsequent social interactions. We examined how preexposure to a loss of control affected social exclusion processing by using the Cyberball paradigm. An event-related brain component (P3) served as a probe for the state of the expectancy system, and self-reports reflected the subjective evaluations of the social threats. In the control group (*n* = 23), the transition to exclusion elicited a significant P3 effect and a high threat to belonging in the self-reports. Both effects were significantly reduced when the exclusion was preceded by preexposure to a loss of control (EG1_disc_, *n* = 23). These effects, however, depend on the offset of the preexposure. In case of a continuation (EG2_cont_, *n* = 24), the P3 effect was further reduced, but the threat to belonging was restored. We conclude that the P3 data are consistent with predictions of a common expectancy violation account, whereas self-reports are supposed to be affected by additional processes.

## 1. Introduction

Social interaction is essential for human beings. Belonging is crucial for psychological well-being [1], whereas social threats reliably trigger negative moods and distress [2]. Being excluded from social participation, such as the experience of being physically isolated from others or being ignored or told one is unwanted, directly impacts individuals’ mental health [3,4,5]. According to Williams’ need–threat model, these consequences are due to threats to fundamental human needs, such as “belonging”, “self-esteem”, “control”, and ”meaningful existence” [6].

The impact of social threats on these needs is typically measured on an established self-report scale, the Need–Threat Questionnaire (NTQ) [7]. Effects in this questionnaire can be triggered reliably by the experience of social exclusion induced in the Cyberball paradigm [6]. This paradigm simulates social exclusion in a virtual ball-tossing game. Participants are instructed to play a ball-tossing game with two putative ”co-players” connected via the Internet to evaluate their visual imagination abilities. The putative ‘co-players’ are computer-generated. The probability of participants’ ball reception is manipulated. Reducing the rate of balls received induces a reliable threat to social needs, which primarily affects the feeling of belonging [8]. This effect was consistently observed in both between-designs [6,9,10,11] and within-designs [12,13,14,15]. The former compares responses of different groups to inclusionary and exclusionary settings, and the latter implies a transition from an inclusionary to an exclusionary setting.

Exclusion, however, is not the only threat in social interaction which induces an immediate aversive response. A comparable response can be elicited if decisional autonomy is challenged by a loss of control that affects the experience of personal control. The social need for control is specifically related to the concept of choice, i.e., the ability to select options [16,17]. Comparable to the threat to belonging, a selective threat to the belief in control will also trigger an aversive state. Evidence has been provided by a study based on a rock–paper–scissors game. It revealed that a loss of control induced by a streak of losses was perceived as annoying and resulted in the highest boredom [18].

Recently, Niedeggen and colleagues [19] modified the Cyberball paradigm, in which personal control was threatened (*Intervention Cyberball*). To challenge the participants’ decisional autonomy, a putative supervisor was introduced, who can overrule the participant’s decision on the recipient of his/her ball throw and select a different ball recipient. Notably, the participant remains included, i.e., the probability of ball reception remains unchanged, but his/her personal control is challenged by the intervention of the supervisor. Correspondingly, self-reports provided in the NTQ indicated that participants did not feel a threat to belonging, but reported a significantly increased threat in the scale need for control [19].

More importantly, the electrophysiological markers induced by a transition to exclusion (*Exclusion Cyberball*) and by a transition to intervention (*Intervention Cyberball*) are comparable, which indicates similarities in the cognitive processing of both social threats. In both paradigms, the most sensitive component identified in the event-related brain potentials (ERPs) is the P3 component. This component refers to a positive voltage deflection at 300–500 ms after the onset of a relevant event that can be evoked in substantial cognitive tasks. In most studies, the P3 component is a more sustained activity defined by a series of peaks related to different stages in attentive and mnestic processing [20]. The frontal component (P3a or frontal P3) has been linked to the early activation of the attentional mechanism [21], whereas the posterior component (P3b or posterior P3) has been related to the process reflecting an update of information in working memory [22].

The posterior P3 effects in Cyberball studies appear to be closely related to processing changes in social participation as well as personal control. In a series of experiments, the change in P3 amplitude was consistently associated with the processing of a transition to exclusion (*Exclusion Cyberball*) [14], or a transition to intervention (*Intervention Cyberball*) [19]. In both cases, the reduced probability of receiving the ball or intervening in decisional autonomy, respectively, increased the P3 amplitude.

It is worth mentioning that the P3 effect induced by a transition to a socially aversive situation is not a mere effect on the change in the probability of a crucial task-relevant event, as previously demonstrated in the established oddball paradigm [23]. The P3 effect obtained in Cyberball studies critically depends on other psychological parameters. For example, the P3 amplitude is reduced if the exclusion is associated with an increase in the number of co-players [14]. Moreover, the P3 amplitude is increased if the avatar of the participant is moved to a superior position [11,15]. In line with earlier ERP studies, it has been concluded that the P3 amplitude rather reflects the violation of experience-based expectations than a change in the probability of a task-relevant event [24,25,26,27].

Following an expectancy-violation account [28], the aforementioned ERP findings obtained in both *Exclusion* and *Intervention Cyberball* reflect an inconsistency between the experience of a social threat and our beliefs or expectations on social participation. This inconsistency is assumed to be associated with aversive affective responses [29]. Several neurocognitive studies suggest that the detections of these inconsistencies rely on the activation of common neural structures [30,31,32] which has been related to an overarching “inconsistency compensation approach” [33]. A common understanding of this approach is that compensation behaviors can be observed for any experience that is inconsistent with participants’ beliefs or goals [33]. Therefore, the inconsistency compensation approach provides a theoretical framework for a more general theory of human behavior [34].

Given that the processing of a loss of control and social exclusion depends on the activation of similar cognitive processes, an interactive effect of both threats is highly likely. The current study was set to explore this interaction by using the P3 component as an index of the expectancy state. Specifically, the purpose of this study is to determine whether the preexposure to a social threat affects the processing of a subsequent aversive situation in the Cyberball game.

The experimental approach shares the idea of social priming, as previously introduced by Hudac [35]. She demonstrated the processing of social exclusion, here induced in a lunchroom task, is modulated by a preceding interpersonal decision. This effect of social priming was also evident in the P3 component. In contrast to Hudac, the current study did not rely on repetitive “prime-target” pairs, but on the transfer effect of an aversive preexposure condition to an exclusion condition. According to the expectancy-violation account [19], preexposure to a loss of control will lead to an adaptation of the expectancy level. This adaptation effect has already been observed in a previous *Intervention Cyberball* study and reflected in a gradual decrease of the P3 amplitude over time [19]. Following the assumptions of a common cognitive system, the adaptation effect driven by a loss of control will also affect the expectation of social involvement, thus influencing the processing of the upcoming exclusion.

In sum, we assumed that expectancies of social participation will be reduced in cases wherein participants have already experienced a loss of control based on the assumption that the expectancy-violation account relies on the activation of a common inconsistency system [19]. Our first specific research question was related to the general effect of “preexposure”. How does pre-exposure to a threat affect the processing of a new threat? To this end, we compared the effects of a transition to exclusion in two groups. In contrast to the controls, the experimental group experienced another threat (here, loss of control) before the onset of exclusion. We hypothesized, that the preexposure to the first threat will affect the expectation for upcoming social threat events. As a result, the P3 effect related to the transition to exclusion should be less pronounced in the experimental group. This effect will also affect the experience of exclusion in this group, and should result in a reduced rating of the threat of belonging in the NTQ. As compared to the other scales (meaningful existence, self-esteem, and control), this scale is more specifically related to the processing of exclusion.

The second specific research question was related to the duration of the preexposure condition: Does the preexposure effect critically rely on the continuation, or discontinuation, of the first threat? Answering this question requires a second experimental group in which the first threat (preexposure) continues and overlaps with the transition to exclusion. The corresponding hypothesis is based on differences in the adjustment of expectancies concerning social interaction in the two experimental groups. In the first experimental group (see above), the offset of the first threat (discontinuation) will probably restore the belief in control and trigger a new adjustment process. The regain of control might therefore reinstate the P3 effect related to the processing of the transition to exclusion, as well as the self-reported threat to belonging. In other words, the positive effects of the offset of the first threat will affect the processing of the following threat. In contrast, the continuation of the first preexposure threat (loss of control) will confirm the preceding adjustment of the expectancy level. According to the general inconsistently model [33], individuals are more prepared for a transition to exclusion and the corresponding markers are less expressed. As compared to the first experimental group, the P3 effect as well as the self-reported threat to belonging are expected to be reduced.

In addition to the a priori defined analysis, we also applied a posteriori data-driven analysis of an additional ERP component, the N2. This component was considered in an a posteriori analysis because it has frequently been observed in Cyberball studies and was related to attentional processes and priming effects [12,35].

## 2. Materials and Methods

The research protocol was approved by the local ethics committee of Freie Universität Berlin (No. 006.2019, 15 May 2019). Before and after the experiment, all participants gave written informed consent according to the Declaration of Helsinki. The study given here was not formally preregistered. Preprocessed EEG data and the source code of the experimental procedure are available here: https://osf.io/9jsbn/?view_only=394a8a7895d14e929168d70b859ef715 (accessed on 2. September, 2022. All manipulations, measurements, and exclusion criteria are reported.

### 2.1. Participants

The sample size was calculated a priori by using G*Power [36] and focused on the sensitivity to detect an interaction effect in a 3 × 2 between–within design (between-participant factor preexposure: not provided, provided but discontinued, provided and continued; within-participant factor ”block”: inclusion block and exclusion block). To probe a medium-sized effect (*f* = 0.20, adjusted to the taxonomy) [37], a sample size of 66 participants was required with a power of 80% by using an *F*-test with alpha at 0.05. The assumption of a medium effect size was based on previous studies, which consistently reported a modulation of the P3 effect in Cyberball studies. Although large effect sizes were reported frequently [14,15,19], a more conservative assumption was made in the novel preexposure setup. The sample of 75 healthy participants was recruited at the FU Berlin and included students from different faculties. Participants already familiar with the Cyberball paradigm were not included. All participants confirmed good physical and mental health. They were randomly assigned to different between-participant conditions (25 participants per condition). Five participants were excluded following a rigorous EEG artifact correction (criteria: see below). The remaining 70 participants were included in the final analysis (51 females; 18–36 years; *M*_age_ = 23.57 years, *SD*_age_ = 4.96 years; all right-handed except 5 participants). Among them, the preexposure provided but discontinued condition, i.e., the first experimental group, comprised 23 participants (14 females; *M*_age_ = 25.30 years, *SD*_age_ = 5.47 years); the preexposure provided and continued condition, i.e., the second experimental group, included 24 participants (20 females; *M*_age_ = 22.79, *SD*_age_ = 3.96); and the condition that did not provide preexposure (control group) involved 23 participants (17 females; *M*_age_ = 22.65 years, *SD*_age_ = 5.10 years). Participants were compensated in credit points or money (20 €).

### 2.2. Task and Design

The setup of the Cyberball game was programmed by using PsychoPy2 software (version: v1.85.6) [38] in Python and followed by established ERP-adjusted designs [19,28]. In addition, controlling the visual display and events management (e.g., the trigger for EEG recording), the PsychoPy2 program was used to record the sequence of ball throws and the corresponding time stamps.

The characteristics of the Cyberball game setting are depicted in Figure 1A. One participant and two putative co-players were represented, respectively, by three avatars on a computer display. The spatial distance among the three avatars was 3° of visual angle. The avatar of the participant was always presented on the computer screen in the inferior and horizontally centered position. To simulate a ball-tossing game, a corresponding ball symbol was presented. The appearance of the ball in spatial proximity to an avatar signals that the player represented by the avatar owns the ball. Participants who held the ball had to pass it to one of two co-players by using the left or right arrow on the keyboard. Pressing the left arrow key indicated that the participant intended to pass a ball to the left co-player while pressing the right arrow key signified that the participant intended to throw a ball to the right co-player. After an arrow key was pressed, the ball vanished for 500 ms and then reappeared beside an intended co-player for 400–1400 ms to enhance the belief of playing with humans.

The experiment consisted of two experimental blocks (see Figure 1B), each containing 200 ball throws. The “inclusion block” (block 1) was defined by equivalent ball reception probability for three players (33% each, i.e., the participant and two co-players received the ball about 66 times, separately). Following this, the “exclusion block” (block 2) was defined by a reduction of ball reception probability for the participant (17%, i.e., the participant only received the ball approximately 34 times). Previous studies have demonstrated that this partial exclusion, which involved a modest reduction in the probability of the ball reception, was sufficient to reliably induce an effect of ostracism [8,12,15,28]. In addition to the manipulation of the probability of participants’ ball reception in the experimental blocks, the design foresees a second manipulation, the intervention or loss of control. In case of an intervention, the recipient of the participant’s ball throw was not the intended co-player, but the remaining co-player (e.g., the left arrow was pressed, but the ball was received by the right co-player). In our experiment, the intervention manipulation was provided in block 1 of the first experimental group and in block 1 and block 2 of the second experimental group. In a block with intervention, the probability of ball reception by the non-intended co-player was 30%. For clarity, we refer to the two experimental groups as follows. The first experimental group will be labeled as EG1_disc_ to indicate the discontinuation of the first threat in block 2, and the second experimental group will be labeled as EG2_cont_ to indicate the continuation of the first threat in block 2.

For all participants, the following chronological sequence of events was applied in the experiment. When the participants arrived at the laboratory, they first provided written informed consent and then were instructed that they would participate in a study measuring visual imagination capabilities and completed a corresponding questionnaire about visual imagination ability (VVIQ) [39]. This questionnaire was designed as a cover story and the data were not further analyzed. After the electrodes were attached, the participants were seated comfortably in front of a computer monitor (7° × 7° of visual angle) at a viewing distance of 120 cm with their jaws on a fixed chin rest with adjustable height. After that, the participants were instructed to play the ball-tossing game.

At the beginning of the Cyberball game, the participants chose one of six avatars to represent themselves in the subsequent game [40] and were told that the other two co-players (connected via the internet) would be also represented by two avatars. Then, the participants were instructed that ball reception was indicated by the ball next to their avatars and that they should press the left or right arrow key to pass the ball to one of two co-players after they received the ball. In the cover story (see above), the participants were also instructed to imagine playing the ball-tossing game on a meadow or at a beach—depending on the experimental block. Moreover, they were informed that a supervisor—who was not a co-player—might intervene in their decision on the recipient of their ball throw and that the supervisor’s activity would vary randomly in different experimental blocks [19]. When the supervisor intervened, the ball was passed to a non-intended instead of the intended recipient.

When the ball-tossing game started, all participants went through a short practice session with inclusion (i.e., participants’ ball reception is 33%) and control (i.e., no intervention) lasting about 4 min (100 ball throws in total). This practice session was designed to ensure that participants’ levels of expectations concerning the involvement and control were set at a comparable level before the experiment started. It was followed by two experimental blocks of about 7 min each (see above for details). Social inclusion was provided in block 1 (mental imagery task: playing on a meadow) and social exclusion was provided in block 2 (mental imagery task: playing at a beach) in all three groups. Meanwhile, the supervisor was active and intervened in block 1 of both EG1_disc_ and EG2_cont_, which indicated that the two experimental groups initiated preexposure to a loss of control. The supervisor’s interventions did not occur in block 2 of EG1_disc_, but continued to occur in block 2 of EG2_cont_ with the same probability as in block 1 (30%).

Immediately following the end of the second experimental block, participants were asked to provide self-reports and to answer a set of questions. This set includes the following three questionnaires: First, the Need–Threat Questionnaire (NTQ). The subscales of the needs–threat scale were reported to be reliable (Cronbach’s αs = 0.71 to 0.79) [41]. In this study, the NTQ consisted of six items related to fundamental social needs in two dimensions (three items per dimension), belonging (e.g., “I felt disconnected”; Cronbach’s α = 0.78) and control (e.g., “I felt I had control over the course of the game”; Cronbach’s α = 0.73). As previous studies have shown, experiencing social exclusion significantly affected the need for belonging [1,8,12], and the increasing frequency of intervention selectively threatened the need for control [19]. We, therefore, renounced the addition of the other NTQ scales (self-esteem, meaningful existence) which are not specifically related to our experimental manipulation. Second, a questionnaire on the self-assigned personal power assessment (e.g., “I felt independent”; with Cronbach’s α = 0.84) [42]. It is important to note that the reduced reliability (Cronbach’s α = 0.55) in this study is due to the restriction to two items. The scale included two items and was used to check whether interventions affecting participants’ decisions violate the expected personal control. The third questionnaire was a negative mood scale (e.g., “I felt bad”). Here, the agreement with eight emotional adjectives was used to measure the negative mood with good reliability (Cronbach’s α = 0.90), which is in agreement with previous studies [43]. All sixteen items above were assessed on a 7-point scale (ranging from −3, “Much Stronger in Block 1” to 3, “Much Stronger in Block 2”) requiring a relative judgment. Participants rated the extent to which their feelings were more expressed in block 1 compared to block 2, or adverse. This relative rating allows us to focus on the changes in feelings from block 1 to block 2, and reduces the effort attached to an independent judgement on the two blocks. In addition, participants were asked to separately estimate the frequency of their ball reception and intervention in block 1 and block 2 (see Appendix A for details).

At the end of the experiment, participants were informed about the real purpose of the study and signed informed consent again.

### 2.3. EEG Recording and Preprocessing

The EEG signals were acquired with a digital amplifier (BrainAmps amplifier, BrainProducts, Gilching, Germany). Eight active Ag/AgCl electrodes (AFz, Fz, F3, F4, Cz, Pz, P7, P8) were embedded in an elastic cap (EASYCAP, Herrsching, Germany). Signals from active EEG electrodes were referenced to linked earlobes. FCz served as a ground electrode. The impedance of all electrodes was kept below 10 kΩ. To monitor blinks and eye movements, vertical electrooculograms (EOGs) were recorded by two electrodes (impedance < 20 kΩ per electrode) placed above and below the right eye, and horizontal EOGs were recorded by two electrodes placed at the outer left and right canthi, respectively. EEG data were recorded continuously at a sample rate of 500 Hz with a 0.1–100 Hz online band-pass filter.

EEG data were preprocessed by using Brain Vision Analyzer (Version: 2.1, Brain Products, Gilching, Germany). Data were offline-filtered (0.3 to 30 Hz, 24 dB/Oct), segmented (epoching: −100 to 800 ms) relative to the onset of ball possession of the participants, and baseline-corrected (−100 to 0 ms). In the next step, trials were automatically identified and excluded from analysis if ocular artifacts (EOG > 50 μV) occurred in a single segment. Epochs with amplitudes exceeding ± 80 μV of active electrodes were marked by using a semi-automatic artifact detection. Furthermore, marked trials were inspected by manual correction for slow movement-related artifacts (drifts) affecting the baseline or high-frequency bursts. We also rejected data from participants whose averaged ERP signal was based on fewer than ten segments per block. Similar exclusion criteria have been applied during previous studies [19,28,44]. Due to these conservative rejection standards, the final averaged ERP in this study relied on a mean of 40.29 trials in block 1 which referred to a rejection rate of 39% (control group: *M* = 47.17 trials, *SD* = 9.50 trials, range 27–65 trials; EG1_disc_: *M* = 44.09 trials, *SD* = 9.79 trials, range 20–62 trials; EG2_cont_: *M* = 30.04 trials, *SD* = 11.88 trials, range 12–53 trials) and on a mean of 19.86 trials in block 2 which referred to a rejection rate of 42% (control group: *M* = 23.17 trials, *SD* = 4.91 trials, range 14–34 trials; EG1_disc_: *M* = 21.04 trials, *SD* = 5.94 trials, range 13–33 trials; EG2_cont_: *M* = 15.54 trials, *SD* = 5.01 trials, range 10–25 trials).

### 2.4. Statistical Analysis

#### 2.4.1. Self-Reported Data

First of all, the estimated frequency of participants’ ball reception was analyzed by running a 3 × 2 ANOVA (SPSS version 27, IBM) to check experimental manipulation, including the between-participant variable (3 levels: control group vs. EG1_disc_ vs. EG2_cont_) and the within-participant variable (2 levels: block 1 vs. block 2). Reported degrees of freedom and *p*-values were corrected according to Greenhouse–Geisser. In case of a significant interaction, post-hoc comparisons were performed.

To test our hypotheses, the analysis focused on the changes on the NTQ scale belonging which reflects the effects of a transition to exclusion—induced in all three groups—as most sensitive. In addition, the analysis was also applied to the self-reports for negative mood. This scale provided insight into the affective changes induced by a threat of social needs [2], but is less specifically related to the processing of exclusion. We analyzed the changes in the self-reported need threat to control. The scale should reflect the effects of an offset of intervention (EG1). Information on the scale of personal power added additional information on the effects of intervention [19]. We used the mean score of each dimension as a measure and higher absolute values indicated larger differences between the two blocks (items were reverse-coded if necessary). The data of these four dimensions (belonging, control, personal power, and negative mood) were analyzed by running a one-way ANOVA with the between-participant variable (control group vs. EG1_disc_ vs. EG2_cont_), respectively. If there was a significant main effect, pairwise post-hoc comparisons were conducted. In addition, a Pearson’s correlation analysis was used to check the correlations between the need threat to belonging and negative mood. Additional self-reports (the estimated intervention frequency) are reported in the supplementary material (see Appendix A).

#### 2.4.2. EEG Data

The hypotheses were focused on the analysis of the P3 effect induced by the transition from inclusion to partial exclusion. To identify the time range in which the P3 amplitude was mostly expressed, we computed the grand-averaged potentials for the event ”ball reception of participants” in block 1 and block 2—independently of the group assignment (see Figure 2C). For the difference wave (block 2–block 1, see Figure 2B) which reflected the P3 effect, the global field power (GFP) was computed. Based on the maximum difference (at approximately 400 ms), we selected two ranges for analysis: the ascending part of the P3 effect (300–400 ms) followed by its sustained activity (400–500 ms). In line with previous studies [20], we assumed that the latter was more related to the classical P3b which was of primary interest concerning our hypotheses. Further analysis of the spatial distribution of the P3 effect revealed significant differences between electrodes (AFz, F3, F4, Fz, Cz, Pz, P7, P8), which was due to the centro-parietal shift of the activity. To account for this effect and avoid multiple tests, two electrode clusters were determined (anterior–frontal: AFz, F3, F4, Fz, centro-parietal: Cz, Pz, P7, P8). Similar approaches to building electrode clusters have been reported in previous ERP studies [28,35,45]. The results Section 3 would focus on the more sensitive centro-parietal P3 effect. Results for the anterior–frontal P3 effect were provided in the supplementary material (see Appendix A).

The mean centro-parietal P3 amplitudes for the windows referring to the “ascending P3” or ”early P3” (300–400 ms) and “descending P3” or “late P3” (400–500 ms) were determined for each participant, which was analyzed by running a 3 (preexposure: control group vs. EG1_disc_ vs. EG2_cont_) × 2 (block: block 1 vs. block 2) ANOVA, respectively. The freedom and *p*-values were also corrected by Greenhouse–Geisser. When interaction occurred, post-hoc tests were run to identify the groups in which the P3 effect induced by the transition to exclusion was significantly expressed.

To test the reliability observed for the analysis of the P3 time ranges, a peak analysis was performed. A peak analysis is usually applied for transient ERP components in which a clear global maximum can be defined in a restricted time range. As already indicated in the grand-averaged potentials (see Figure 2c). These prerequisites hardly hold for the P3 component obtained in the single experimental blocks. The P3 is a more sustained component, and in most participants the time range of interest (300–500 ms) is defined by a series of local maxima. Therefore, a reliable detection of a P3 peak in the ERPs related to each block was not possible. Therefore, peak detection was based on the P3 difference wave elicited by the transition to exclusion (block 2–block 1, see Figure 2B). By using the time windows defined above, we ran an automatic peak analysis at centro-parietal sites and found that the positive maxima in single participants can be identified in the time range of 300–500 ms. The results of the peak detection were further manually controlled. Data, in which the detection was not possible, were marked and excluded from further analyses. In the case of valid data (expression of a positive maximum in the pre-defined time range), latency and baseline-related amplitude of the P3 peak were included in the consequent statistical analysis, respectively. Valid data were identified in the data of the participants of the control group and EG1_disc_. However, the P3 was only detected in 12 out of 24 datasets in the group EG2_cont_. Therefore, we restricted the P3 peak analysis to the control group and EG1_disc_. The latency and amplitude were analyzed by running a one-way ANOVA with two levels—the control group and EG1_disc_, respectively.

The analysis of the EEG/ERP data also involved two exploratory analyses: The first analysis was driven by the visual inspection of the grand-averaged ERP in EG1_disc_. We observed transient negativity in centro-parietal leads at about 180 ms (N2). This N2 component plays a crucial role in attention and social priming which has been also reported in earlier Cyberball-EEG studies [12,35]. Based on its temporal and spatial characteristics, the mean amplitude in the time range 140–220 ms was computed and pooled for the electrodes (Cz and Pz). The mean amplitude was also analyzed by running a 3 (preexposure: control vs. EG1_disc_ vs. EG2_cont_) × 2 (block: block 1 vs. block 2) ANOVA. Post-hoc comparisons would be motivated by a significant interaction of the experimental factors.

In the second analysis, we explored the frontal alpha asymmetry. In previous studies [13,46], the transition to exclusion was associated with a shift in EEG signal, and this shift has been related to an altered approach/withdrawal motivation. In contrast to the previous studies, this EEG parameter did not respond to our experimental manipulations. The details of the analysis and the results can be found in the supplementary material (see Appendix A) or in the data repository https://osf.io/9jsbn/?view_only=394a8a7895d14e929168d70b859ef715 (accessed on: 2 September 2022).

## 3. Results

### 3.1. Self-Report

#### 3.1.1. Manipulation Check

In all three experimental groups, participants noticed that they received the ball less frequently after the transition from the inclusion to the exclusion block (see Table 1). The ANOVA confirmed the significant main effect of block, *F*(1, 67) = 157.314, *p* < 0.001, *η*_p_^2^ = 0.701. The main effect of preexposure, *F*(2, 67) = 0.901, *p* = 0.411, *η*_p_^2^ = 0.026, was not significant, and the interaction was also not significant, *F*(2, 67) = 0.902, *p* = 0.411, *η*_p_^2^ = 0.026. This indicated that the reduction of ball reception was recognized reliably.

#### 3.1.2. Questionnaires (NTQ and Negative Mood)

As the description presented in Table 1 and Figure 2A, the two social needs of interest and negative emotions were affected by a transition from inclusion to exclusion. However, the changes were differentially expressed due to preexposure in three groups.

As for the threat to belonging, changes were expressed in the control group and EG2_cont_ but not in EG1_disc_. Please note that a negative value reflects that the threat to belonging was stronger in block 2 as compared to block 1, i.e., it was increased with the transition to exclusion. The ANOVA confirmed a significant difference between three groups, *F*(2, 67) = 6.257, *p* = 0.003, *η*_p_^2^ = 0.157. The post-hoc analyses revealed that compared to the control group, the threat of belonging in EG1_disc_ was decreased significantly, *F*(1, 44) = 5.015, *p* = 0.03, *η*_p_^2^ = 0.102. There was no significant difference in the threat to belonging between the EG2_cont_ and the control group, *F*(1, 45) = 1.567, *p* = 0.217, *η*_p_^2^ = 0.034. When comparing the two experimental groups, the threat to belonging was significantly higher in EG2_cont_ as compared to EG1_disc_, *F*(1, 45) = 11.336, *p* = 0.002, *η*_p_^2^ = 0.201.

As for the threat to need for control, the transition to exclusion did not trigger a remarkable change in each group. A negative value would signal that the threat to control is higher expressed in block 2 when compared to block 1, and vice versa. Although mean values—referring to the differential rating block 2–block 1—were slightly increased in the control group and EG2_cont_, the ANOVA did not indicate a significant difference between the three groups, *F*(2, 67) = 2.614, *p* = 0.081, *η*_p_^2^ = 0.072.

The descriptive effect provided in the scale control was more expressed for the rating of personal power. Table 1 shows that the self-reported personal power was reduced by a transition to exclusion (block 2 < block 1) in Control and EG2, whereas it was slightly enhanced in EG1. A group-specific difference was confirmed by the ANOVA, *F*(2, 67) = 6.447, *p* = 0.003, *η*_p_^2^ = 0.161. In line with the descriptive pattern, the rating of personal power was restored in EG1 as compared to the control group, *F*(1, 44) = 9.712, *p* = 0.003, *η*_p_^2^ = 0.181, as well as compared to EG2, *F*(1, 45) = 8.283, *p* = 0.006, *η*_p_^2^ = 0.155. We did not observe a difference in the personal power ratings between the control group and EG1, *F*(1, 45) = 0.033, *p* = 0.857, *η*_p_^2^ = 0.001.

As for the scale negative mood, the pattern was comparable to the result for the belonging scale. A high positive value—referring to an increase of negative mood in block 2 when compared to block 1—was found in the control group and EG2_cont_ but not in EG1_disc_. In other words, the transition to exclusion triggered a negative mood in these groups. A group-specific effect was confirmed by the ANOVA, *F*(2, 67) = 6.603, *p* = 0.002, *η*_p_^2^ = 0.165. Compared to the control group, negative mood was significantly decreased in EG1_disc_, *F*(1, 44) = 9.639, *p* = 0.003, *η*_p_^2^ = 0.180, but not changed in EG2_cont_, *F*(1, 45) = 0.002, *p* = 0.965, *η*_p_^2^ = 0.000. A direct comparison of EG1_disc_ and EG2_cont_ indeed revealed a significant difference, *F*(1, 45) = 8.856, *p* = 0.005, *η*_p_^2^ = 0.164.

Given that, to a certain extent, we observed a similar pattern in the need threat to belonging and in negative mood with the transition from inclusion to exclusion in the three groups, we computed intercorrelations in the next step. Pearson’s correlation analysis indicated a strong correlation between the threat to belonging and negative mood in three groups, *r* = −0.740, *p* < 0.001, and in each group (control group, *r* = −0.675, *p* < 0.001; EG1_disc_, *r* = −0.766, *p* < 0.001; EG2_cont_, *r* = −0.640, *p* = 0.001).

### 3.2. ERP Results

Descriptive statistics on ERP data are presented in Table 1. In addition, Figure 2C depicts the grand-averaged ERP separated into the anterior and posterior electrode clusters in the three groups. To indicate the effect of a transition to exclusion, ERP traces referring to block 1 and block 2 are superimposed in the illustration. Additionally, the strength of the ERP effect is pointed out in Figure 2B presenting the difference wave (block 2–block 1).

As shown in Figure 2C, the ERP response to participants’ ball reception is characterized by a common pattern in all groups and conditions. Transient negativity (N2) and a sustained P3 response occurred. The N2 component, at about 180 ms, was increased visibly at posterior leads in EG1_disc_ if ball reception probability was reduced, but it didn’t change much in the control group and EG2_cont_. In terms of the P3 component, the first local maximum was found at about 300 ms, followed by a global maximum at about 400 ms. This maximum corresponded to the maximum of the GFP and was more expressed at centro-parietal sites. As described above (see Methods Section 2), the analysis of the experimental effects was focused on the centro-parietal P3 effect. Based on the GFP, we defined two temporal ranges (300–400 ms and 400–500 ms) referring to the ascending and descending part of the P3. To validate statistical findings in the time ranges, we also determined the P3 peaks in the control group and EG1_disc_ and analyzed latencies and baseline-to-peak amplitudes (As mentioned above, this analysis does not include EG2_cont_). A posteriori, the effects on the mean amplitudes of centro-parietal N2 (Cz and Pz, 140–220 ms) were analyzed. Please note that the results of the analysis of the fronto-central clusters can be found in the Appendix A.

#### 3.2.1. Early P3 Amplitude (300–400 ms)

In the ascending part of the P3, there was an apparent P3 effect with the transition to exclusion in the control group, but the corresponding effects were reduced in EG1_disc_ and not expressed in EG2_cont_. The ANOVA indicated a significant main effect of “block”, *F*(1, 67) = 8.32, *p* = 0.005, *η*_p_^2^ = 0.110, and a significant interaction effect of block and preexposure on the P3 amplitude, *F*(2, 67) = 3.424, *p* = 0.038, *η*_p_^2^ = 0.093, but no significant main effect of preexposure, *F*(2, 67) = 1.402, *p* = 0.253, *η*_p_^2^ = 0.040. The post-hoc comparisons confirmed that these results were due to the difference in P3 effect (block 2–block 1, see Figure 2B) between the control group and EG2_cont_, *F*(1, 45) = 5.797, *p* = 0.020, *η*_p_^2^ = 0.114. No significant differences in the P3 amplitude were found between EG1_disc_ and the control group, *F*(1, 44) = 1.850, *p* = 0.181, *η*_p_^2^ = 0.040, as well as between EG1_disc_ and EG2_cont_, *F*(1, 45) = 1.939, *p* = 0.171, *η*_p_^2^ = 0.041.

#### 3.2.2. Late P3 Amplitude (400–500 ms)

The effect observed in the early part of the P3 was more expressed in the succeeding time range. As shown in Figure 2B, the P3 effect remained the strongest in the control group, weakened in EG1_disc_, and was not expressed much in EG2_cont_. This pattern was confirmed by the ANOVA resulting in a significant main effect of block, *F*(1, 67) = 15.608, *p* < 0.001, *η*_p_^2^ = 0.189, and a significant interaction effect of block and preexposure, *F*(2, 67) = 5.151, *p* = 0.008, *η*_p_^2^ = 0.133, but again no significant main effect of preexposure, *F*(2, 67) = 2.227, *p* = 0.116, *η*_p_^2^ = 0.062. The post-hoc comparison revealed that the differences in the P3 effect were significantly reduced in both experimental groups when compared to the control group: control vs. EG2_cont_: *F*(1, 45) = 8.135, *p* = 0.007, *η*_p_^2^ = 0.153; control vs. EG1_disc_, *F*(1, 44) = 4.243, *p* = 0.045, *η*_p_^2^ = 0.088. Differences in the P3 effect were not significant between EG1_disc_ and EG2_cont_, *F*(1, 45) = 2.076, *p* = 0.157, *η*_p_^2^ = 0.044.

#### 3.2.3. P3 Peak (300–500 ms)

The P3 peaks were determined to confirm the results of the time ranges mentioned above. Please consider that a peak analysis for the data of EG2_cont_ was not possible: As depicted in Figure 2B, the P3 was not reliably expressed in this group (see methods). As indicated in Table 1, the peak amplitude of the P3 effect was significantly decreased in EG1_disc_ as compared to the control group, *F*(1, 44) = 4.551, *p* = 0.039, *η*_p_^2^ = 0.094. The latencies of the P3 peak were determined at about 420 ms in the two groups, confirming that the P3 effect was rather expressed in the late part of the P3 component. The ANOVA confirmed that the estimated latencies did not differ significantly between EG1_disc_ and the control group, *F*(1, 44) = 0.027, *p* = 0.871, *η*_p_^2^ = 0.001.

#### 3.2.4. N2 Amplitude (140–220 ms)

Based on the visual inspection of the grand-averaged ERP, the N2 effect was considered for analysis a posteriori. Here, the transition to exclusion led to an increase of the N2 mean amplitude exclusively in EG1_disc_ (see Figure 2C). The ANOVA showed that there was a significant interaction of the block factor and preexposure, *F*(2, 67) = 3.72, *p* = 0.029, *η*_p_^2^ = 0.100, but no significant main effects of block, *F*(1, 67) = 2.027, *p* = 0.159, *η*_p_^2^ = 0.029 or preexposure, *F*(2, 67) = 1.963, *p* = 0.148, *η*_p_^2^ = 0.055. The post-hoc analysis confirmed a significant difference of the N2 effect between the EG1_disc_ and the control group, *F*(1, 44) = 8.484, *p* = 0.006, *η*_p_^2^ = 0.162. The remaining pairwise comparisons were not significant: EG1_disc_ vs. EG2_cont_: *F*(1, 45) = 3.701, *p* = 0.061, *η*_p_^2^ = 0.076; control vs. EG2_cont_: *F*(1, 45) = 0.266, *p* = 0.609, *η*_p_^2^ = 0.006.

## 4. Discussion

This study tested whether a formerly experienced loss of control affected the processing and evaluation of a subsequent social exclusion. In line with our hypotheses, the preexposure to one social threat (loss of control) affected the participants’ reactions to another social threat (i.e., the transition to exclusion), specifically the expression of the event-related brain potentials (ERPs, e.g., P3 component) as well as the expression of the threat to belonging provided in the self-reports. However, the expression of the P3 effect and the perceived threat to belonging depended critically on the continuation of the first threat. As compared to the control group (no preexposure), discontinuation of the first threat (EG1_disc_) led to a reduction of the P3 effect and the threat to belonging. In the case of continued loss of control (EG2_cont_), the P3 effect was even further reduced, but the threat to belonging was restored. Together, these results support the notion that the experience of a loss of control influences the subsequent experience of social exclusion and that the continuation/discontinuation of the first threat differentially modulates the effects of the second threat. ERP results and self-reported results will be discussed separately in the following.

### 4.1. Effects of preexposure on ERPs

The ERP results confirm the idea that the concept of “priming” can be transferred to the processing of social interactions [35]. In the present study, the transfer effect is clearly expressed for the posterior P3 component, which serves as a probe of the expectancy state in the Cyberball paradigm [14,28]. According to the previous findings, the P3 effect triggered by the transition to exclusion reflects the deviance from the expectancy level of social involvement. ERP responses to the event “ball reception” serve as a probe because this event evokes stimulus evaluation in the context of the preceding information. A similar process has been proposed to explain the trial-by-trial fluctuation of the P3 amplitude [47].

In the present study, the P3 effect associated with a transition to exclusion was modulated by the previous threat of intervention (loss of control). In other words, the magnitude of the P3 effect—as measured in the Cyberball game—was not only influenced by the subjective probability to receive the ball [48] but, more importantly, depended on the previous experience of another unrelated threat. A similar transfer process has previously been observed for social threat learning which modified the behavior in a subsequent decision-making task [49].

The P3 effects obtained in the current study reveal that preexposure to a different social threat modulates the expectation for social participation: The aforementioned re-adjustment process triggered by the transition to exclusion was expressed in the control group, but was less expressed in the experimental groups. This reflects that less—or no—adjustment of the subjective expectation is necessary for the preexposure groups. Notably, the reduction of the P3 effects was modulated by the continuation of the first threat: Whereas there was no evidence for a readjustment of expectancies if the preexposure threat was continued (EG2_cont_), the P3 effect was observed in case of an offset of the preexposure threat (EG1_disc_).

These effects can be related to a recent model on expectancy violation (ViolEx 2.0 model) [50]. According to the model, individuals respond to expectation violation with accommodation or immunization. The former refers to an update of expectations to promote consistency with the experienced event, whereas the latter refers to the avoidance of an update to shield a priori expectations. Importantly, the degree of the expectancy violation—reflected in the P3 effect—is assumed to depend on an internal anticipatory reaction that affects the way how an upcoming aversive situational outcome will be processed [51]. Internal anticipatory reactions aim to facilitate adaptive accommodation processes to expectancy violations. We assume that these processes share the characteristics of a readjustment process and are triggered by preexposure to the first threat. When individuals encounter exclusion events in the second block, the internal anticipatory reaction reduces the magnitude of expectation violation.

If the preexposure threat continues (EG2_cont_), the internal anticipatory reaction is confirmed, and the magnitude of the expectancy violation induced by the onset of the exclusion is reduced. This effect is indicated by the prominent reduction of the P3 effect.

If the preexposure threat is discontinued (EG1_disc_), the participants also experience a regain of control. We suppose that this process primarily weakens the validity of the internal negative expectation. According to the model, a destabilization of expectation is probably triggered by the offset of interventions, which may provide a pleasant violation of expectation [52]. This destabilization might increase the magnitude of the expectancy violation induced by the onset of the new threat (social exclusion). As a consequence, the amplitude of the P3 effect is increased. We tentatively propose that the reduced validity of the internal representation also provoked an earlier increasing central-parietal N2 effect—exclusively observed in EG1_disc_. This ERP component has been related to conflict monitoring/detection processes and the redistribution of attentional resources [53,54]. Both processes have been linked to the internal anticipatory reactions [50], which affect effective attention guidance and the processing of prediction error signals [55].

### 4.2. Effects of preexposure in self-reports 

Based on previous findings, we assumed that a reduction of the P3 effect should be associated with a corresponding reduction in the self-reported threats to social needs in Cyberball games [12,13,14,28,56]. This relation, however, was only confirmed in EG1_disc_, but not in EG2_cont_. In EG2_cont_, the threat to belonging induced by a transition to exclusion was significantly expressed despite the striking reduction of the P3 effect.

These results provide further evidence that responses in the self-reports cannot be predicted reliably on the basis of ERP effects. Earlier studies already indicated that the interindividual variance in the self-reports cannot be sufficiently explained by the expression of the P3 effect [57]. This has been related to fundamental differences between physiological and behavioral measures. The P3 amplitude provides an online probe of cognitive processes closely linked to the adjustment of subjective expectancies. In contrast, an a posteriori evaluation of the aversive experience is additionally affected by affective and motivational processes [13]. In the recent study, we also have to consider that participants were instructed to provide relative judgments (block 1–block 2) in the self-reports.

Based on these considerations, we suggest that the expressions in self-reports are primarily determined by the change in the psycho-affective state induced by the two threats. With block 1 serving as a baseline for the post-hoc evaluation, the transition to exclusion induced an unpleasant expectancy violation in the control group as well as in EG2_cont_. The negative mood induced by the transition led to a bias in the rating on the threat scales (belonging and control), i.e., the shift in mood influenced the rating behavior of participants. In EG1_disc_, the transition to exclusion was accompanied by an offset of the first threat which also reflects a regaining of control. In contrast to the simultaneous transition to exclusion, the regaining of control can be defined as a pleasant expectancy violation. This might reduce the a posteriori evaluation of the negative mood induced by the transition to exclusion, and the rating of the social threats is less affected. The close relationship between the psycho-affective state and the NTQ scales is supported by our correlation analysis (see above). Moreover, previous studies have indicated that negative affective states enhanced pessimistic expectations about other events [58,59].

### 4.3. Limitations 

Several factors limit the conclusions that may be drawn from our experimental results. First, our analysis was primarily focused on the P3 effect elicited by transition to exclusion and linked to the event “ball reception of the participant”. Due to this restriction in ERP analysis, we cannot rule out that other components related to other cognitive and affective processes are engaged in the modulation of the processing of exclusion [13]. These processes are probably more closely linked to self-reports. Secondly, we only indirectly infer that the onset of intervention in block 1 (EG1_disc_ and EG2_cont_) threatens the need for control. The self-reports provided are relative judgments and focus on the change from block 1 to block 2. However, results from previous studies already demonstrated that such interventions provide a reliable threat to control [19]. Thirdly, our results are markedly different from previous results obtained in a social priming experiment [35]. This may partially be attributed to the difference in the experimental task (Cyberball vs. Lunchroom task) and setup (preexposure period vs. prime-target-pairs). Hence, it would be worthwhile to validate the expectation-violation account in a Lunchroom task in future research. Fourth, the effect of the preexposure period was only demonstrated in one direction (here, a loss of control affects expectancy for exclusion). If the assumption of a common cognitive mechanism is correct, these effects should replicate if the order of conditions is swapped. Evidence for this hypothesis will have to be provided in a further study. Furthermore, the composition of the sample reduces the generalizability of our findings. Specifically, the majority of the sample was recruited from a set of undergraduate students. These participants can be related to a Western, well-educated, industrialized, rich, and democratic (WEIRD) social background, which limits general conclusions of the results. The sample was mainly recruited from a set of undergraduate students. We cannot rule out that the expected social participation is different in other samples, and that social threat to belonging and/or control will be evaluated differently. A final limitation can be related to the sample size. Although we detected an interaction of strong effect size in the P3 amplitude, the generalization of the effect would require a much larger sample. In addition, a more heterogeneous sample would enable us to explore potential moderators of the reported effects, such as age or educational background. Recent studies already suggest gender differences in the processing of social pain [60].

## 5. Conclusions

The current study provides evidence that a preexposure to a specific social threat (here, loss of control) affects the processing of a different social threat (here, social exclusion). The ERP results are in line with the predictions of overarching models [33,50] supposing that the processing of aversive events is accompanied with an adjustment of subjective expectations. Most importantly, our data provide evidence that this adjustment process is not specifically related to a social threat, but apparently affects a wide range of subjective beliefs on social interactions. Self-reports, on the other hand, are more likely to reflect a posteriori processes closely linked to the negative effect induced by an aversive social situation.

The idea that an exposure to a specific social threat affects our general beliefs on upcoming social participation appears to be relevant for current developmental [61] and clinical research [62], and will have to be confirmed in further experimental studies.

## Figures and Tables

**Figure 1 brainsci-12-01225-f001:**
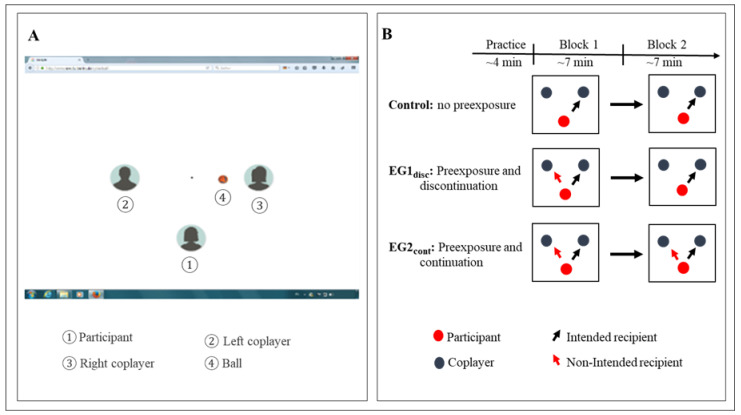
(**A**) Setup of the visual display. The participant and the two putatively connected co-players are represented as avatars. The participant’s avatar was always at a central position. (**B**) Experimental design. Pictograms refer to the experimental condition in the first and second block of the Cyberball game. The control group remained control in the first block and was excluded in the second block. EG1_disc_: In addition to the transition to exclusion, participants experienced a loss of control exclusively in block 1 (discontinued preexposure). EG2_cont_: In addition to the transition to exclusion, participants experienced a loss of control in block 1 and lock 2 (continued preexposure).

**Figure 2 brainsci-12-01225-f002:**
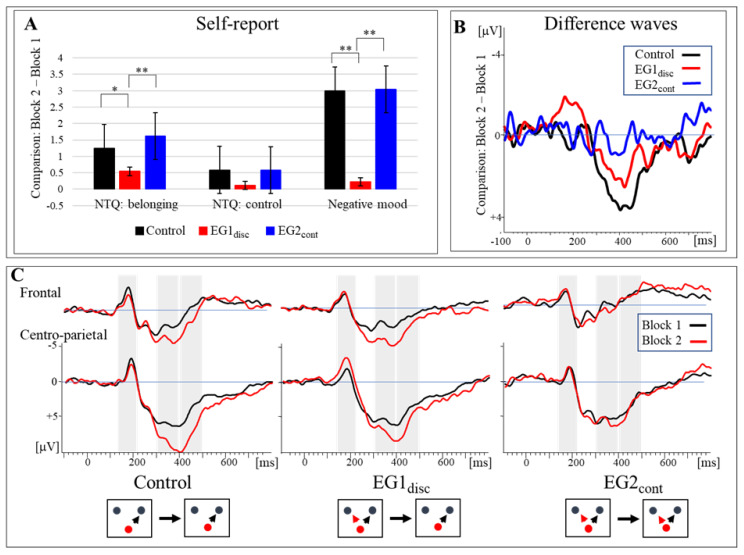
(**A**) Self-report data. The retrospective rating of need threats (NTQ belonging, NTQ control) and negative mood is based on the relative judgement (block 2–block 1). (**B**) Difference ERP waves (block 2–block 1) computed for the centro-parietal electrode cluster: The P3 effect induced by the transition to exclusion is clearly expressed in the controls and EG1_disc_, but not in EG2_cont_. (**C**) Grand-averaged ERP waveforms (block 1 and block 2) separated for the three groups and two experimental blocks. Time regions which are analyzed are shaded: N2 (140–220 ms) and P3 (300–400 ms and 400–500 ms). The P3 effect—induced by the transition to exclusion—is mostly expressed in controls and EG1_disc_ at the central parietal cluster. The N2 amplitude was selectively increased at posterior electrodes in EG1_disc_. Here, the icons refer to the experimental condition (see Figure 1B): The red dot refers to the avatar of the participant. A red arrow indicates that interventions are possible in the experimental block. Control: the control group; EG1_disc_: the first experimental group (the preexposure provided but discontinued); EG2_cont_: the second experimental group (the preexposure provided and continued). * *p* < 0.05; ** *p* < 0.01.

**Table 1 brainsci-12-01225-t001:** Descriptive statistics of the self-report (lines 1–8) and ERP data (lines 9–16). To each mean, the upper and lower confidence intervals (95%) are provided. The negative signs of NTQ mean values indicate that participants feel more threatened in block 2 than in block 1. Mean ERP data are provided for the centro-parietal cluster (P3: Cz, Pz, P7, and P8; N2: Cz and Pz).

	Control (*n* = 23)		EG1_disc_ (*n* = 23)		EG2_cont_ (*n* = 24)	
	*M (SD)*	*CI*	*M (SD)*	*CI*	*M (SD)*	*CI*
NTQ: belonging	−1.25 (0.98)	[−1.67, −0.83]	−0.54 (1.17)	[−1.04, −0.03]	−1.61 (1.02)	[−2.04, −1.18]
NTQ: control	−0.58 (0.75)	[−0.90, −0.26]	−0.12 (0.95)	[−0.53, 0.30]	−0.58 (0.68)	[−0.87, −0.30]
Negative mood	3.00 (2.92)	[1.74, 4.26]	−0.22 (4.02)	[−1.96, 1.52]	3.04 (3.48)	[1.57, 4.51]
Personal power	−0.61 (0.83)	[−0.97, −0.25]	0.28 (1.10)	[−0.19, 0.76]	−0.56 (0.91)	[−0.95, −0.18]
Estimated ball reception in block 1 (%)	35.30 (8.60)	[31.43, 39.18]	32.70 (9.09)	[28.82, 36.47]	32.42 (10.11)	[28.63, 36.21]
Estimated ball reception in block 2 (%)	17.39 (10.85)	[14.08, 20.71]	18.83 (6.56)	[15.51, 22.14]	15.25 (5.58)	[12.01, 18.50]
Estimated intervention in block 1 (%)	8.43 (10.88)	[3.73, 13,14]	23.96 (18.83)	[15.81, 32.10]	18.71 (13.86)	[12.86, 24.56]
Estimated intervention in block 2 (%)	22.35 (26.12)	[11.05, 33.64]	19.70 (22.08)	[10.15, 29.25]	33.13 (21.86)	[23.89, 42.36]
P3 amplitude (300–400 ms) in block 1 (μV)	5.54 (2.57)	[4.55, 6.53]	4.93 (1.99)	[3.94, 5.92]	5.57 (2.54)	[4.60, 6.54]
P3 amplitude (300–400 ms) in block 2 (μV)	7.27 (2.27)	[6.15, 8.40]	5.93 (2.54)	[4.81, 7.05]	5.42 (3.17)	[4.32, 6.51]
P3 amplitude (400–500 ms) in block 1 (μV)	4.06 (2.05)	[3.02, 5.09]	3.62 (2.29)	[2.61, 4.64]	3.86 (2.89)	[2.85, 4.87]
P3 amplitude (400–500 ms) in block 2 (μV)	6.49 (2.88)	[5.28, 7.69]	4.75 (3.17)	[3.55, 5.96]	3.91 (2.61)	[2.73, 5.09]
N2 amplitude (140–220 ms) in block1 (μV)	−0.37 (2.11)	[−1.27, 0.53]	0.09 (2.02)	[−0.81, 0.99]	0.61 (2.33)	[−0.27, 0.1.49]
N2 amplitude (140–220 ms) in block2 (μV)	−0.07 (2.78)	[−1.15, 1.01]	−1.30 (2.18)	[−2.38, −0.22]	0.57 (2.76)	[−0.49, 1.62]
P3 peak amplitude of differences waves (μV)	4.82 (2.13)	[3.90, 5.74]	3.50 (2.06)	[2.61, 4.39]	N/A (N/A)	N/A
P3 latency of differences waves (ms)	419.48 (48.72)	[398.41, 440.54]	417.04 (52.53)	[394.33, 439.76]	N/A (N/A)	N/A

Notes. Control, the control group; EG1_disc_, the first experimental group (the preexposure provided but discontinued); EG2_cont_, the second experimental group (the preexposure provided and continued); *M*, mean; *SD*, standard deviation; *CI*, 95% confidence interval; NTQ, the need threat questionnaire; N/A, not applicable.

## Data Availability

Preprocessed EEG data and the source code of the experimental procedure are available here: https://osf.io/9jsbn/?view_only=394a8a7895d14e929168d70b859ef715 (accessed on: 2 September 2022).

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
