# Peer review of "From Loss of Control to Social Exclusion: ERP Effects of Preexposure to a Social Threat in the Cyberball Paradigm"

_brainsci, 2022, doi:10.3390/brainsci12091225_

Round 1

Reviewer 1 Report

Comments and Suggestions for Authors

This is an interesting study examining how preexposure to a loss-of-control influence social exclusion processing using the Cyberball paradigm. The paper is well-written. The experimental design looks good. I really appreciate the authors effort to upload their data in OSF, which help to increase the credibility of the paper. I believe that the paper may contribute to the literature well. I only have a few comments to improve the manuscript further:

1. It is unclear how the power analysis was conducted. It is reasonable to expect a medium effect size? Perhaps, it is important for the authors to provide more justification on their analytical choice.

2. More information about the participants' demographic will be helpful

3. The authors should report the reliability of the self-reported questionnaires (e.g., NTQ)

4. One of the biggest limitation of the current study is the small sample size. This should be acknowledged in the discussion section. The authors should also discuss how the sample size may affect the result and its interpretation.

Reviewer 2 Report

Comments and Suggestions for Authors

This manuscript investigates how preexposure to a loss-of-control influenced social exclusion processing using the Cyberball paradigm. The ERP component (P3) served as a probe for the expectation of social participation, and self-reports reflected the subjective evaluations of the social threats. They had three groups: control (n=23) where subjects experienced a typical inclusion-exclusion Cyberball game; EG1 (n=24) where subjects pre-exposed to a loss-of-control before the exclusion; and EG2 (n=24) where subjects pre-exposed to a loss-of-control and continued to be in a loss-of-control while experiencing the exclusion. The P3 effect (increased P3 amplitude from inclusion to exclusion context) and a high threat to ‘belonging’ in the self-reports were significantly reduced when the exclusion was preceded by preexposure to a loss-of-control (and discontinued) (EG1). In case of a continuation of loss-of-control (EG2), the P3 effect was further reduced but the threat to ‘belonging’ was restored. The authors concluded that the P3 data are consistent with predictions of a common expectancy violation account, whereas self-reports are supposed to be affected by additional processes such as the negative affect induced by an aversive social situation.

Overall, this study offers insight into the role of pre-exposure to one type of social threat in social exclusion processing. The study setup (especially hypotheses) is somewhat unclear and there are also some important omissions with respect to the measurements, reported results, and some aspects need further discussion.

1.    Hypotheses 1 and 2 should be more clearly stated. Since the outcomes are the same between these two hypotheses and the fundamental differences between hypotheses 1 and 2 are: H1) difference between EG1=EG2 vs Control, and H2) difference between EG1 vs EG2 vs Control, I do not see the point in having Hypothesis 1. Hypothesis 2 should be able to answer the question if the general effect of ‘preexposure’ to the threat to ‘belongings.’ Or does hypothesis 2 not require a comparison with the control group? It would be better to state how the authors planned to compare those three groups for each hypothesis.

2.    Were there no hypotheses related to the threat to need for ‘control’ and negative mood? What was the motivation to measure ‘personal power”? Those should be clearly stated in the introduction and in the methods.

3.    L246. The original NTQ measures the perceived level of social need threat with four subscales: belonging, self-esteem, meaningful existence, and control). It was not clear why the authors only used two of them. Was a negative mood scale taken from the NTQ or was this study original since they changed the rating system? What kind of rating system was used for NTQ and other scales (e.g., ranging from -3 ‘Much Stronger in Block 1’ to 3 ‘Much Stronger in Block 2’; or x-point scales ranging from 1 (not at all) to x (very much))? Which scales were measured as relative changes between block 1 and block 2?

4.    Line 342. “The grand averages already indicated that a reliable peak detection within the sustained P3 wave was hardly possible.” Can the authors elaborate on this sentence?  

5.    Line 347. “The results of the peak detection were further manually controlled: Data, in which a positive deflection was not identified in the predefined time range, were marked.” Marked and then how was it further processed?

6.    L397. Results of negative mood. It was hard to understand which group showed reduced or increased negative mood from inclusion to exclusion. This should be clearly stated with statistics instead of a descriptive statement.

7.    The motivations for two exploratory analyses in the EEG data (N2 component and alpha asymmetry) were not introduced in the Introduction and it was out of the blue in Lines 356-370.

8.    Table 1. Why is estimated intervention in block 2 so high in the Control (22.35%)?

9.    Why were P3 effects not consistent with the results of threat to ‘belonging’? P3 was an index of expectation of social participation and it would be associated with threat to ‘belonging.’

10. Subjects in the EG1disc not only experienced discontinuation of loss-of-control, but rather they experienced regaining control over the social choice (EG1disc) at block 2 (social exclusion). Thus the exclusion-inclusion effect is probably confounded with the sense of regaining control. I see the authors did the analysis related to a self-report of ‘personal power’ which indicates EG1 showed increased personal power from block 1 to block 2 (supplemental Table S1). The statistics do not look appropriate though (EG2 vs Control showed a significant difference instead of EG1 vs Control).

The sense of personal power could be related to negative mood and other social factors such as threat to 'belonging' and 'control,' which might have hindered the inclusion-exclusion effects.  

11. How did the authors deal with non-intervention trials and intervention trials? Did they average ERP across those trials? There could be a different ERP manifestation between those trials.

12. Why were post-hoc analyses done with F-statistics instead of t-statistics? Did the authors add any covariates (e.g., age, sex)? Did they control multiple testing?

minor:

It would be helpful if the authors could provide Appendix (questionnaire used in the study) in English.

Round 2

Reviewer 1 Report

Comments and Suggestions for Authors

The authors have addressed all my comments well. I appreciate all their efforts.